# Measurement accuracy and cutoffs for predicting primary aldosteronism diagnosis using Lumipulse® for renin and aldosterone measurements

**Maude Laney**[1], **Appoline Nabi-Bourgois**[1], **Julien Mallart-Riancho**[2,3,4,5], **Simon Travers**[1,3,4], **Mouna Chebbi**[1], **Aurore Kling**[1], **Laurence Amar**[2,3,4,5], **Stephanie Baron**[1,3,5,6]*

**1** Physiology Department, AP-HP, Georges Pompidou European Hospital, Paris, France, **2** Hypertension Unit, AP-HP, Georges Pompidou European Hospital, Paris, France, **3** Université Paris Cité, Paris, France, **4** Inserm, Paris Cardiovascular Research Center, Paris, France, **5** Adrenal Referral Center, Université Paris Cité, Paris, France, **6** Inserm, CNRS, Cordeliers Research Center, Université Paris Cité, Sorbonne Université, Paris, France

* stephanie.baron@aphp.fr

## Abstract

### Objectives

Primary aldosteronism (PA) diagnosis is a multistep process that begins with screening based on the plasma aldosterone-to-renin ratio (ARR). As cut-off values depend on the methods used, the aim of this study was 1) to validate the performances of new aldosterone and renin chemiluminescent immuno-assays using the Lumipulse® analyzer (Fujirebio®) and then 2) to define the criteria to distinguish PA patients from those with essential hypertension (EH).

### Methods

This monocentric retrospective study included 297 patients admitted to Georges Pompidou European Hospital, France, between January 2021 and December 2022 for the assessment of hypertension etiology. We evaluated analytical performances and compared Lumipulse® results for plasma renin (PRC) and plasma aldosterone (PAC) concentrations (n = 196) as well as 24-hour urine aldosterone excretion (24UA) (n = 201) to the reference methods (PRC: Liaison XL® Diasorin®; PAC and 24UA: LC-MS/MS). Receiver operating characteristic curve analysis was used to define the cut-off values for ARR, PAC, and 24UA to distinguish between PA and EH patients.

### Results

Analytical characteristics were in accordance with requested performances. Our results showed excellent correlations between Lumipulse® and our reference methods. We propose cut-off values of 35 pmol/mIU as the ARR, 260 pmol/L for PAC, and 49 nmol/24h for 24UA for predicting PA diagnosis.

**Data availability statement:** All relevant data are within the paper and its Supporting Information files.

**Funding:** Fujirebio Europe sponsored this study, but did not participate in data analysis nor in writing the manuscript.

**Competing interests:** The authors have declared that no competing interests exist.

## Conclusions

Lumipulse® (Fujirebio®) is a reliable alternative for measuring PRC, PAC, and 24UA for predicting PA diagnosis with the use of corresponding cut-off values for each criterion.

## Introduction

Primary aldosteronism (PA) is the one of the most common causes of secondary hypertension and is reported to be diagnosed in 6% to 20% of hypertensive patients [1]. Compared with essential hypertension (EH), PA is associated with increased target organ damage and cardio-vascular morbidity [2–5], thus stressing the need for an early and easily accessible diagnosis [6]. Clinical practice guidelines for PA diagnosis involve a multistep procedure, beginning with the measurement of plasma renin (PRC) and aldosterone (PAC) concentrations to compute the aldosterone-to-renin ratio (ARR) in association with basal PAC levels [7] with correctly measured low PRC [8]. Numerous studies suggest using various cut-off values for ARR and PAC depending on the type of PAC and PRC assays [9–11]. In addition, 24-hour urine aldosterone excretion (24UA) is a useful tool to confirm hyperaldosteronism [12,13].

Accurate and reproducible measurements of renin and aldosterone are therefore essential, with several methods being widely used. Regarding the plasma and urine aldosterone measurements, LC-MS/MS assays are now considered gold standard. Nonetheless, LC-MS/MS methods are not widely available, require specific technical expertise, and cannot be used in all laboratories. In this way, automated inmuno-assays represent the most efficient alternative for aldosterone and renin measurements.

Fujirebio® recently developed fully automated electro-chemiluminescent immuno-assays with the Lumipulse® analyzer (Tokyo, Japan) for PRC, PAC, and 24UA measurements. Both renin and, more interestingly, aldosterone measurements are based on the sandwich method, which could improve the specificity of aldosterone measurements compared with routine aldosterone immuno-assays. A few studies have evaluated this method [14–18], although none of them defined specific cut-off values for PA prediction using Lumipulse® renin and aldoste-rone measurements.

The aim of this study was therefore 1) to compare Lumipulse® with the previously vali-dated immuno-assay method for PRC and the gold standard LC-MS/MS method for PAC and 24UA, and 2) to define cut-off values for the prediction of PA diagnosis for ARR, PAC, and 24UA using Lumipulse®.

## Materials and methods

### Study design and participants

This monocentric retrospective study included two cohorts of patients admitted to Georges Pompidou European Hospital between January 2021 and April 2022 for the assessment of hypertension etiology. For each patient, their full medical history, complete physical examina-tion, and laboratory tests were recorded. Office blood pressure was determined based on the average of three measurements after a 5-minute rest [19]. EH was diagnosed after extensive evaluations, including biological tests and adrenal and renal artery imaging to exclude sec-ondary hypertension according to 2023 guidelines [20]. PA diagnosis was made in accordance with international guidelines [7].

Between January 2021 and April 2022, 2,125 patients were screened. Medical records were accessed between April 2022 and January 2024. Included patients were aged above 18 years with a complete medical record. Exclusion criteria were pregnant or breastfeeding women and

those with an incomplete medical record. For blood assessments specifically, only patients for whom PRC and PAC were sampled between 7 and 9 am in standard semi-sitting position, with sufficient ethylenediaminetetraacetic acid (EDTA) plasma for both PRC and PAC measurements on were included. Samples from included patients did not undergo any freeze thaw cycle prior to Lumipulse® PRC, PAC or 24UA measurements. Thus, for the evaluation of PRC and PAC, 196 patients were included in the final analysis.

For the evaluation of 24-hour urine aldosterone excretion specifically, only patients for whom 24-hour urine collection was complete were included. Thus, for 24UA, 201 patients were included in the final analysis. Samples and medical records were pseudonymized. This study was registered with the local institutional review board #00001072.

**Routine methods.** For the measurement of PAC and PRC, blood samples were collected in EDTA-K3 between 8h and 9h after a 30-minute rest in the seated position [21] with the aim to compute ARR. PRC routine measurements were performed with XL® (Diasorin®, Saluggia, Italy) calibrated against NISBC MRC 68/356 [22,23]. PAC was measured using LC-MS/MS with a Xevo TQS Waters® system (Milford, Mass, USA) [9]. An ARR higher than 46 pmol/mIU on at least two different occasions and a baseline PAC in the seated position higher than 360 pmol/L on at least two different occasions confirmed the diagnosis of PA [9]. If PRC was less than 5 mIU/L, the 5 mIU/L value was used to calculate ARR [24,25]. A confirmatory saline infusion test was performed in all patients except those with contraindications (severe symptomatic hypertension, severe hypokalemia, or congestive heart failure) and those with PAC above 550 pmol/L and PRC below quantification level (<2 mIU/L) [7]. Aldosterone secretion was considered non-suppressible if the post-SIT PAC was higher than 139 pmol/L [7]. 24UA was measured using LC-MS/MS with a Xevo TQS Waters® system after acid hydrolysis. Hyperaldosteronism was confirmed if 24UA was higher than 65 nmol/24h. The completeness of 24-hour urine collection was checked in accordance with criteria for 24-hour urine creatinine excretion [26]. Potassium, sodium, and creatinine plasma as well as urine concentrations were determined using standard methods (AU5800®, Beckman Coulter®, Villepinte, France). Adrenal venous sampling (AVS) was performed to distinguish between unilateral and bilateral aldosterone secretion. Patients with lateralized aldosterone secretion who underwent adrenalectomy returned 6 months after surgery for the post-operative evaluation of PA biochemical cure (ARR, PAC, and kalemia), according to the primary aldosteronism surgical outcome (PASO) criteria [27].

**PRC, PAC, and 24UA measurements with Lumipulse®.** Aliquots of EDTA-K3 plasma and 24-hour urine collection were stored for less than 18 months at -80°C between sampling and Lumipulse® measurements (Fujirebio®, Tokyo, Japan). Required sample volumes for aldosterone and PRC were 40 μL and 50 μL, respectively. All measurements were performed blindly in accordance with manufacturer instructions. Blood samples from 196 patients sampled in the standard seated position were included for PRC and PAC evaluations. Lumipulse® PRC was expressed in mIU/L with conversion factor 1 mIU/L = 0.49 pg/mL. Among these 196 samples, 105 were from EH patients and 52 from PA patients. To determine the PA cut-off values, only samples from patients without drugs interfering with the renin-angiotensin-aldosterone system (RAAS) were used (i.e., n = 44 PA and n = 59 EH). Beta-blockers, angiotensin II type 1 receptor blockers, angiotensin-converting enzyme inhibitors and diuretics were discontinued 2 weeks before blood collection; mineralocorticoid receptor antagonists and renin inhibitors 6 weeks before. Calcium antagonists, α-blockers, and centrally acting antihypertensive drugs were prescribed to patients with severe hypertension if necessary.

Complete 24-hour urine collections from 201 patients were included for 24UA evaluation. Lumipulse® 24UA measurements were performed after the overnight acid hydrolysis of urine

samples as recommended by the manufacturer. Among these 201 samples, 99 came from EH patients and 35 from PA patients. For PA diagnosis, only samples from patients without drugs interfering with RAAS (as described previously) were included (i.e., n = 32 PA and n = 62 EH).

Imprecision studies were conducted in accordance with the CLSI EP15 protocol [28]. Intra-assay variation was determined by analyzing the 30 repetitions of the same sample on the same day for low, medium, and high concentration pools for each parameter. Inter-assay variations were determined by analyzing three quality controls for PRC, two for PAC, and two samples of urine for 24UA in each separate batch.

Method comparison was performed for PRC, PAC, and 24UA values obtained using Lumipulse® Fujirebio® compared with previously validated routine methods. This study was conducted according to the tenets of the Declaration of Helsinki.

## Statistical analysis

Intra- and inter-assay variability were expressed as mean +/- standard deviation. Bland–Altman analysis (bias and limits of agreement), Passing–Bablok regression, and correlation tests were used for method comparison for PRC, PAC, and 24UA.

Basal characteristics of the population were compared using a parametric t-student test after verifying the normal distribution. Receiver-operator curve (ROC) with area under the curve (AUC) calculations, likelihood ratios, Youden tests, and positive (PPV) and negative predictive values (NPV) were used to define sensitivity, specificity, and cut-off values for ARR, PAC, and 24UA. Results were expressed as mean with 95% confidence interval (95% CI) unless otherwise specified. $p$ values below 0.05 were considered significant. Data analysis was performed with Prism® (La Jolla, California, USA), version 8.4 for Windows.

# Results

## Analytical performances

Analytical performances of PRC, PAC, and 24UA are summarized in S1 Table.

**Plasma renin.** The intra-assay coefficient variation (CV) ranged from 1.8% to 8.6%. Inter-assay CV ranged from 3.5% to 6.7% for the different concentration levels. Lumipulse® analytical performances were as good as those for the reference method using Liaison XL®. Samples used to compare the PRC methods (n = 131 samples with PRC above the low limit of quantification for both methods) between Liaison XL® and Lumipulse® ranged from 2.7 to 1002 mIU/L and from 3.0 to 820 mIU/L, respectively. Passing–Bablok regression showed a relation: $PRC_{Lumipulse} = 0.797$ (95 CI: 0.771;0.82) x $PRC_{Liaison\ XL} - 0.375$ (95 CI: -0.866; 0.162) (Fig 1A). Bland-Altman analysis showed a mean bias of –7.6 mIU/L and a 95% limit of agreement between –47.6 and +32.5 mIU/L (Fig 1B).

**Plasma aldosterone.** Intra-assay CV ranged from 0.7% to 2.4%, whereas inter-assay CV was below 3.5% regardless of the concentration. Lumipulse® analytical performances were as good as those for reference LC-MS/MS method. Samples used for the PAC method comparison between LC-MS/MS and Lumipulse® ranged from 44 to 3521 pmol/L and from 30 to 3127 pmol/L, respectively (n = 195). Passing–Bablok regression showed a relation: $PAC_{Lumipulse} = 0.951$ (95 CI: 0.927; 0.980) x $PAC_{LC-MS/MS} - 13.688$ (95 CI: -20.035; - 7.371) (Fig 1C). Bland-Altman analysis showed a mean bias of –33.0 pmol/L and a 95% limit of agreement between –165 and + 99 pmol/L (Fig 1D).

**Urine aldosterone.** Intra-assay CV was below 1.7% regardless of the level, while inter-assay CV was below 5.5%. Lumipulse® analytical performances were as good as those for reference LC-MS/MS method. Sample concentrations to compare the urine aldosterone method between LC-MS/MS® and Lumipulse® ranged from 1.4 to 206.2 nmol/L and from

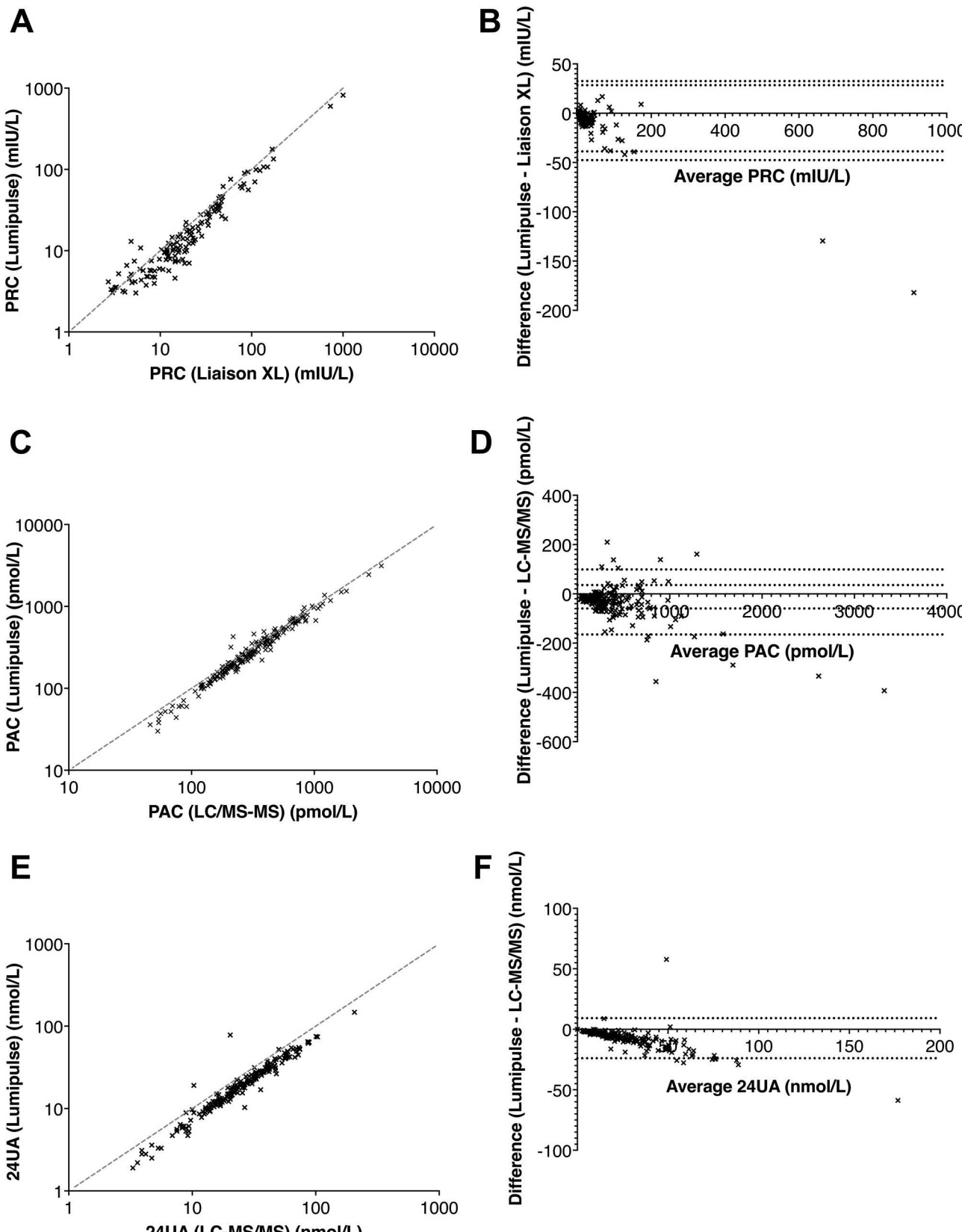

**Fig 1. Plasma renin method comparison between Lumipulse® (Fujirebio®) and Liaison XL® (Diasorin®); A: correlation, B: Bland-Altman (Lumipulse® – Liaison XL).** Plasma aldosterone method comparison between Lumipulse® (Fujirebio®) and LC-MS/MS; C: correlation, D: Bland-Altman

(Lumipulse® – LC-MS/MS). 24-hour urine aldosterone method comparison between Lumipulse® (Fujirebio®) and LC-MS/MS; E: correlation, F: Bland-Altman analysis (Lumipulse® – LC-MS/MS).

1.9 to 147.4 nmol/L, respectively (n = 198). Passing–Bablok regression showed a relation: $UA_{Lumipulse}$ = 0.752 (95 CI: 0.735; 0.770) x $UA_{LC-MS/MS}$ – 0.197 (95 CI: -0.524; 0.088) (Fig 1E). Bland-Altman analysis showed a mean bias of –7.4 nmol/L and a 95% limit of agreement between –23.95 and + 9.18 nmol/L (Fig 1F).

## Determining new cut-off values for predicting primary aldosteronism diagnosis

**Description of the blood sampling population.** Among the 196 patients, 52 had PA, 105 EH, and 39 other miscellaneous diseases (Fig 2). Among the 52 PA and 105 EH patients, 44 and 59, respectively, did not take RAAS-interfering drugs. All 44 PA patients underwent AVS. Among them, 18 exhibited aldosterone lateralized secretion at AVS, with 11 of them undergoing an adrenalectomy. Among these 11 PA patients, 7 had a post-operative examination, which showed a corresponding complete biochemical cure based on the primary aldosteronism surgical outcome criteria, 1 patient exhibited a partial biochemical cure, and 3 patients were lost to follow-up. Among the 7 other patients, 5 of them did not have follow up in our center, one passed away prior adrenalectomy and the last one finally refused surgery.

The demographic and clinical characteristics of patients are summarized in Table 1. Males represent respectively 55.9% of EH patients and 56.8% of PA patients. Age, sex, body mass index (BMI), systolic blood pressure (SBP), diastolic blood pressure (DBP), CKD-EPI eGFR, and 24-hour urine sodium excretion were not significantly different between EH and PA patients. Overall, 75% of PA patients received potassium supplementation compared with only 22% of EH patients. Potassium supplementation was systematically prescribed when

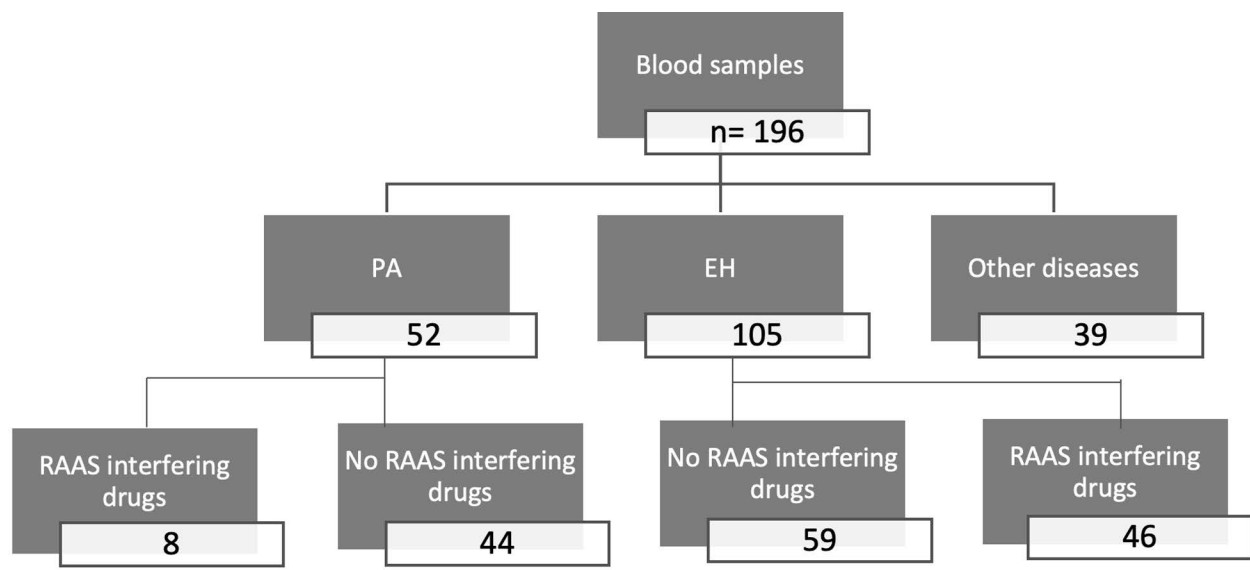

**Fig 2. Flow chart for the evaluation of primary aldosteronism blood criteria.** Patients were categorized into three groups: primary aldosteronism (PA), essential hypertension (EH), and other miscellaneous diseases. All samples were used for the method comparison, while only samples from PA and EH patients without renin-angiotensin-aldosterone system-interfering drugs were used for determining the PA cut-off.

**Table 1. Demographic characteristics of patients included for determining the ARR and PAC cut-off values.**

| | Essential Hypertension | Primary aldosteronism | *p* value |
|---|---|---|---|
| *n* individuals (% male) | 59 (55.9%) | 44 (56.8%) | >0.9999 |
| Age (years) | 40.0 [37.7–43.5] | 43.4 [40.4–46.4] | 0.1878 |
| BMI (kg/m²) | 28.8 [27.3–30.3] | 28.3 [26.9–29.8] | 0.6906 |
| Systolic blood pressure (mmHg) | 144 [139 – 149] | 148 [143–154] | 0.3099 |
| Diastolic blood pressure (mmHg) | 88 [84 –92] | 93 [89–97] | 0.1140 |
| Potassium supplementation (%) | 22% | 75% | <0.0001 |
| Plasma potassium (mmol/L) | 3.9 [3.8–4.0] | 3.6 [3.5–3.8] | 0.0018 |
| eGFR CKD-EPI (ml/min/1.73 m²) | 95.1 [90.8–99.5] | 96.4 [90.6–102.1] | 0.7228 |
| PRC (mIU/L) | 14.6 [10.1–19.1] | 3.0 [1.9–4.0] | <0.0001 |
| PAC (pmol/L) | 272 [229–315] | 661 [463–859] | <0.0001 |
| ARR (pmol/mIU) | 29.4 [23.0–36.1] | 127.9 [88.0–167.6] | <0.0001 |
| 24-h urine sodium excretion (mmol/24h) | 129 [114–145] | 142 [122–162] | 0.2378 |
| 24-h urine potassium excretion (mmol/24h) | 56 [49–62] | 89 [72–105] | 0.0002 |
| 24-hour urine aldosterone (nmol/24h) | 44.1 [36.9–51.1] | 133.4 [68.0–199.0] | 0.0025 |

Data are expressed as mean [95% confidence interval] or (percentage). BMI: body mass index; eGFR: estimated glomerular filtration rate; PRC: plasma renin concentration; PAC: plasma aldosterone concentration; ARR: aldosterone-to-renin ratio.

patients exhibited any history of hypokalemia in accordance with guidelines [7]. Compared with EH patients, PA patients had a lower plasma potassium concentration (p = 0.0018) and PRC ($p < 0.0001$) and a higher PAC, ARR, 24UA, and 24-hour urine potassium excretion ($p < 0.0001$).

**Description of the 24-hour urine aldosterone excretion population.** Among the 201 patients, 35 had PA, 99 EH, and 67 other miscellaneous diseases. Among the 35 PA and 99 EH patients, 32 and 62, respectively, did not take RAAS-interfering drugs (Fig 3). Among these 32 PA patients, 27 underwent AVS, with 9 of them undergoing a lateralized aldosterone secretion. Among these 9 patients, 5 underwent an adrenalectomy with 4 post-operative PA biochemical cures and 1 patient without follow-up.

The demographic and clinical characteristics of the population are summarized in Table 2. Males represent respectively 51.6% of EH patients and 59.4% of PA patients. Age, sex, BMI, and SBP did not significantly differ between EH and PA patients. Overall, 71.9% of PA patients received potassium supplementation compared with only 16.1% of EH patients. PA patients had a lower PRC (p = 0.0005), plasma potassium concentration (p = 0.0151), and CKD-EPI eGFR (p = 0.0277) and a higher DBP (p = 0.0039), PAC, ARR, 24UA, and 24-hour urine potassium excretion ($p < 0.0001$).

**ARR cut-off for PA detection using Lumipulse®.** ARR was computed using 5 mIU/L if Lumipulse® PRC results were below 5 mIU/L, a cut-off value used for ARR computation based on the current recommendations [13].The CI% 5th–95th percentile of mean measured with Lumipulse® for seated ARR was 81.0 to 151.8 pmol/mIU and 26.0 to 38.2 pmol/mIU in PA and EH patients, respectively (Fig 4A).

ARR ROC curve analysis was performed for EH (n = 59) and PA patients (n = 44). AUC was 0.8841 (95% CI: 0.8225–0.9456) (p < 0.0001) (Fig 4B). For the maximal Youden index (63.25), the ARR cut-off value was 35.2 pmol/mIU and yielded a sensitivity of 95.5% (95% CI: 84.9–99.2) and specificity of 67.8% (95% CI: 55.1–78.3) for PA detection (Table 3). The positive likelihood ratio (LR) for this cut-off was 2.96 with PPV of 24.8% and NPV of 99.3%. A cut off value of 31.4 pmol/mIU would reach 97.7% sensitivity and 57.6% specificity. To reach

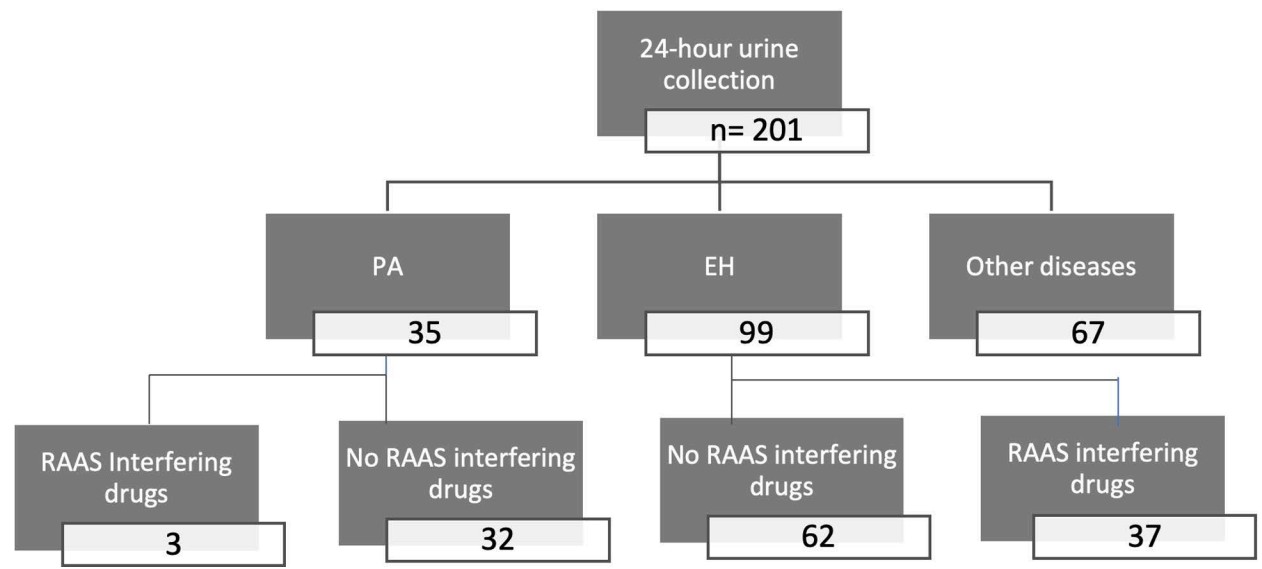

**Fig 3. Flow chart for the evaluation of primary aldosteronism 24-hour urine aldosterone excretion.** Patients were categorized into three groups: primary aldosteronism (PA), essential hypertension (EH), and other miscellaneous diseases. All samples were used for the method comparison, while only samples from PA and EH patients without renin-angiotensin-aldosterone system-interfering drugs were used for determining the PA cut-off.

**Table 2. Demographic characteristics of patients included for determining the 24-hour urine aldosterone excretion cut-off values.**

|  | Essential Hypertension | Primary aldosteronism | *p* value |
|---|---|---|---|
| n individuals (% male) | 62 (51.6%) | 32 (59.4%) | 0.5181 |
| Age (years) | 42.1 [39.5–44.7] | 45.9 [43.1–48.8] | 0.0695 |
| BMI (kg/m²) | 28.6 [26. –29.6] | 29.3 [27.4 – 31.2] | 0.3602 |
| Systolic blood pressure (mmHg) | 144 [139–150] | 152 [145–159] | 0.0765 |
| Diastolic blood pressure (mmHg) | 86 [82–91] | 97 [92–103] | 0.0039 |
| Potassium supplementation (%) | 16% | 72% | <0.0001 |
| Plasma potassium (mmol/L) | 3.9 [3.8–4.0] | 3.7 [3.6–3.8] | 0.0151 |
| eGFR CKD-EPI (ml/min/1.73 m²) | 97.7 [93.7–101.8] | 88.9 [81.0–96.7] | 0.0277 |
| PRC (mIU/L) | 13.6 [9.2–18.0] | 2.3 [1.6–3.1] | 0.0005 |
| PAC (pmol/L) | 286 [245–327] | 554 [433–675] | <0.0001 |
| ARR (pmol/mIU) | 34.1 [27.5–41.1] | 107.9 [83.8–131.8] | <0.0001 |
| 24-h urine sodium excretion (mmol/24h) | 117 [104–131] | 142 [123–162] | 0.0331 |
| 24-h urine potassium excretion (mmol/24h) | 60 [53–67] | 88 [74–102] | <0.0001 |
| 24-h urine aldosterone (nmol/24h) | 45.8 [39.4–52.2] | 74.4 [59.7–88.8] | <0.0001 |

Data are expressed as mean [95% confidence interval] or (percentage). BMI: body mass index; eGFR: estimated glomerular filtration rate; PRC: plasma renin concentration; PAC: plasma aldosterone concentration; ARR: aldosterone-to-renin ratio.

100% sensitivity, corresponding cut off value for ARR would be 19.7 pmol/mIU with 37.3% specificity.

**PAC cut-off for PA detection using Lumipulse®.** The CI% 5th–95th percentile of mean measured with Lumipulse® for seated PAC in PA and EH patients was 446 to 799 pmol/L and 207 to 283 pmol/L, respectively (Fig 4C). PAC ROC curve analysis was performed for EH

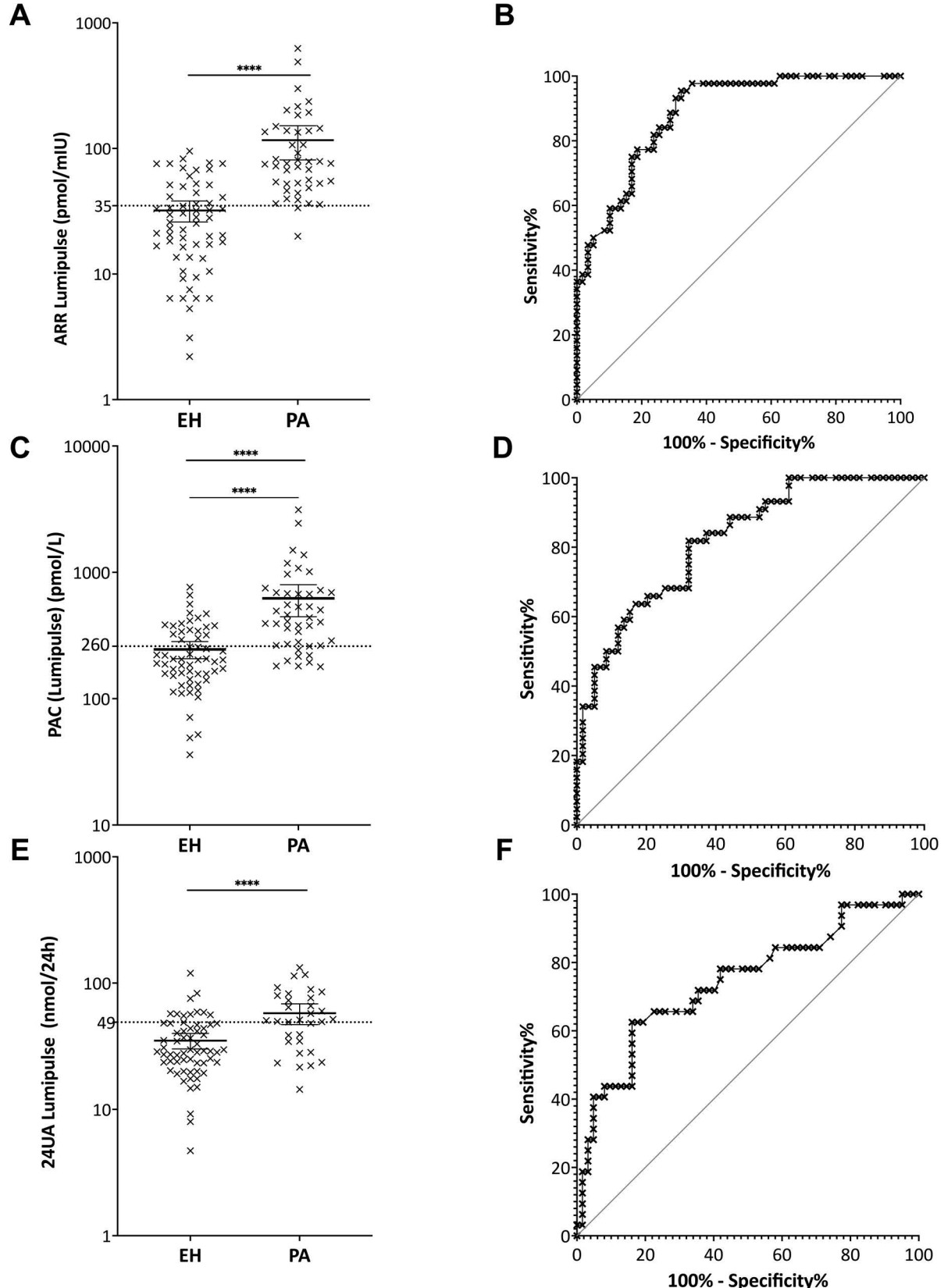

**Fig 4. Primary aldosteronism cut-off definition using Lumipulse® for the aldosterone-to-renin ratio (ARR).** A: Primary aldosteronism (PA) and essential hypertension (EH) ARR distribution, B: ARR ROC curve; for plasma aldosterone concentration (PAC), C: PA and EH

PAC distribution, D: PAC ROC curve; for 24-hour urine aldosterone excretion (24UA), E: PA and EH 24UA distribution, F: 24UA ROC curve.

**Table 3. Performance of aldosterone-to-renin ratio (ARR) cut-off values for primary aldosteronism detection: primary aldosteronism *versus* essential hypertension patients using Lumipulse®.**

| ARR (pmol/mIU) | Sensitivity (%) [95% CI] | Specificity (%) [95% CI] | Youden index | LR + | LR- | PPV | NPV |
|---|---|---|---|---|---|---|---|
| > 19.7 | 100 [92.0–100] | 37.3 [26.1–50.1] | 37.29 | 1.60 | 0.00 | 0.15 | 1.00 |
| > 22.7 | 97.7 [88.2–99.9] | 45.8 [33.7–58.3] | 43.49 | 1.80 | 0.05 | 0.17 | 0.99 |
| > 28.3 | 97.7 [88.2–99.9] | 52.5 [40.0–64.7] | 50.27 | 2.06 | 0.04 | 0.19 | 0.99 |
| > 31.4 | 97.7 [88.2–99.9] | 57.6 [44.9–69.4] | 55.36 | 2.31 | 0.04 | 0.20 | 0.99 |
| > 35.2 | 95.5 [84.9–99.2] | 67.8 [55.1–78.3] | 63.25 | 2.96 | 0.07 | 0.25 | 0.99 |
| > 36.4 | 93.2 [81.8–97.7] | 69.5 [56.9–79.8] | 62.67 | 3.05 | 0.10 | 0.25 | 0.99 |
| > 39.4 | 88.4 [73.3–93.6] | 71.2 [58.6–81.2] | 57.55 | 3.00 | 0.19 | 0.25 | 0.98 |
| > 42.7 | 84.0 [70.6–92.1] | 74.6 [62.2–83.9] | 58.67 | 3.31 | 0.21 | 0.27 | 0.98 |
| > 45.2 | 81.8 [68.0–90.5] | 76.2 [64.0–85.3] | 58.09 | 3.45 | 0.24 | 0.28 | 0.97 |
| > 47.5 | 79.6 [65.5–88.9] | 76.3 [64.0–85.3] | 55.82 | 3.35 | 0.27 | 0.27 | 0.97 |
| > 55.5 | 65.9 [51.1–78.1] | 83.1 [71.5–90.5] | 48.96 | 3.89 | 0.41 | 0.30 | 0.96 |
| > 70.8 | 59.1 [44.4–72.3] | 89.8 [79.5–95.3] | 48.92 | 5.81 | 0.46 | 0.39 | 0.95 |
| > 82.4 | 38.6 [25.7–53.4] | 96.6 [88.5–99.4] | 35.25 | 11.40 | 0.64 | 0.56 | 0.93 |
| > 101.0 | 36.4 [23.8–51.1] | 100 [93.9–100] | 36.36 | | 0.64 | 1.00 | 0.93 |

CI: confidence interval; LR +: positive likelihood ratio; LR-: negative likelihood ratio; NPV: negative predictive value; PPV: positive predictive value.

(n = 59) and PA patients (n = 44) and exhibited an AUC of 0.8211 (95% CI: 0.7428–0.9001) (p < 0.0001) (Fig 4D). For the maximal Youden index (49.62), the PAC cut-off value was 258 pmol/L and yielded a sensitivity of 81.8% (95% CI: 68.0–90.5) and specificity of 67.8% (95% CI: 55.1–78.3) for PA detection (Table 4). The positive LR for this cut-off was 2.541 with PPV of 22.0% and NPV of 97.1%. To reach 90% sensitivity, PAC cut off value would be 197 pmol/L with 47.5% specificity; while to reach 90% specificity the cut off value would be 450 pmol/L with 47.7% sensitivity.

**24UA cut-off value for PA detection using Lumipulse®.** The CI% 5th–95th percentile of mean for 24UA measured with Lumipulse® in PA (n = 32) and EH (n = 62) patients was 46.9 to 68.5 nmol/24h and 30.1 to 40.0 nmol/24h, respectively (Fig 4E).

24UA ROC curve analysis performed between EH and PA patients exhibited an AUC of 0.7419 (95% CI: 0.310–0.8529) (p < 0.001) (Fig 4F). A 24UA cut-off value of 49.1 nmol/24h yielded a sensitivity of 62.5% (95% CI: 45.3–77.1) and specificity of 83.9% (95% CI: 72.8–91.0) for PA detection (Table 5). The positive LR for this cut-off was 1.031, whereas the Youden index was 46.37, PPV 30.1%, and NPV 95.3%. A 90% sensitivity would be reached with a cutoff value of 22.8 nmol/L24h with 22.6% specificity; a 90% specificity would be reached with cut off value of 57 nmol/24h with 43.8% sensitivity.

## Discussion

PA is the first cause of secondary hypertension. The diagnosis of PA begins with screening that requires the determination of PRC and PAC to compute ARR [7,29–31], with a potential confirmatory step if needed. Nonetheless, this diagnosis is difficult to confirm, mostly because of the fluctuating aldosterone secretion despite the standardized conditions to perform ARR

**Table 4. Performance of basal plasma aldosterone concentration (PAC) cut-off values for primary aldosteronism detection: primary aldosteronism *versus* essential hypertension patients using Lumipulse®.**

| PAC (pmol/L) | Sensitivity (%) [95% CI] | Specificity (%) [95% CI] | Youden index | LR + | LR- | PPV | NPV |
|---|---|---|---|---|---|---|---|
| > 178 | 100 [92.0–100] | 39.0 [27.6–51.7] | 38.98 | 1.64 | 0.00 | 0.15 | 1.00 |
| > 197 | 90.9 [78.8–96.4] | 47.5 [35.3–60.0] | 38.37 | 1.73 | 0.19 | 0.16 | 0.98 |
| > 219 | 86.4 [73.3–93.6] | 57.6 [44.9–69.4] | 43.99 | 2.04 | 0.24 | 0.18 | 0.97 |
| > 258 | 81.8 [68.0–90.5] | 67.8 [55.1–78.3] | 49.62 | 2.54 | 0.27 | 0.22 | 0.97 |
| > 266 | 75 [60.6–85.4] | 67.8 [55.1–78.3] | 42.8 | 2.33 | 0.37 | 0.21 | 0.96 |
| > 275 | 72.7 [58.2–83.7] | 67.8 [55.1–78.3] | 40.53 | 2.26 | 0.40 | 0.20 | 0.96 |
| > 339 | 68.2 [53.4–80.0] | 76.3 [64.0–85.3] | 44.45 | 2.87 | 0.42 | 0.24 | 0.96 |
| > 389 | 59.1 [44.4–72.3] | 86.4 [75.5 – 93.0] | 45.53 | 4.36 | 0.47 | 0.33 | 0.95 |
| > 450 | 47.7 [33.8–62.1] | 91.5 [81.7–96.3] | 39.26 | 5.64 | 0.57 | 0.39 | 0.94 |
| > 500 | 43.2 [29.7–57.8] | 94.9 [86.1–98.6] | 38.1 | 8.50 | 0.60 | 0.49 | 0.94 |
| > 558 | 34.1 [21.9–48.9] | 94.9 [86.1–98.6] | 29.01 | 6.71 | 0.69 | 0.43 | 0.93 |
| > 691 | 25 [14.6–39.4] | 98.3 [91.0–99.9] | 23.31 | 14.79 | 0.76 | 0.62 | 0.92 |
| > 866 | 18.2 [9.5 – 32.0] | 100 [93.9–100] | 18.18 | | 0.82 | 1.00 | 0.92 |

CI: confidence interval; LR +: positive likelihood ratio; LR-: negative likelihood ratio; NPV: negative predictive value; PPV: positive predictive value.

**Table 5. Performance of 24-hour urine aldosterone excretion (24UA) cut-off values for primary aldosteronism detection: primary aldosteronism *versus* essential hypertension patients using Lumipulse®.**

| 24UA (nmol/24h) | Sensitivity (%) [95% CI] | Specificity (%) [95% CI] | Youden index | LR + | LR- | PPV | NPV |
|---|---|---|---|---|---|---|---|
| > 11.9 | 100 [89.3–100] | 4.8 [1.3–13.3] | 4.839 | 1.05 | 0.00 | 0.11 | 1.00 |
| > 21.1 | 96.9 [84.3–99.8] | 22.6 [14.0–34.4] | 19.46 | 1.25 | 0.14 | 0.12 | 0.99 |
| > 21.9 | 93.8 [79.9–98.9] | 22.6 [14.0–34.4] | 16.33 | 1.21 | 0.28 | 0.12 | 0.97 |
| > 22.8 | 90.6 [75.8–96.8] | 22.6 [14.0–34.4] | 13.21 | 1.17 | 0.42 | 0.12 | 0.96 |
| > 23.6 | 87.5 [71.9–95.0] | 25.8 [16.6–37.9] | 13.31 | 1.18 | 0.48 | 0.12 | 0.95 |
| > 28.0 | 81.3 [64.7–91.1] | 43.6 [31.9–55.9] | 24.8 | 1.44 | 0.43 | 0.14 | 0.95 |
| > 33.9 | 78.1 [61.3–89.0] | 58.1 [45.7–69.5] | 36.19 | 1.86 | 0.38 | 0.17 | 0.96 |
| > 34.7 | 75 [57.9–86.8] | 58.1 [45.7–69.5] | 33.06 | 1.79 | 0.43 | 0.17 | 0.95 |
| > 37.7 | 71.9 [54.6–84.4] | 64.5 [52.1–75.3] | 36.4 | 2.03 | 0.44 | 0.18 | 0.95 |
| > 47.5 | 65.6 [48.3–79.6] | 77.4 [65.6–86.0] | 43.05 | 2.91 | 0.44 | 0.24 | 0.95 |
| > 50.8 | 50 [33.6–66.4] | 83.9 [72.8–91.0] | 33.87 | 3.10 | 0.60 | 0.26 | 0.94 |
| > 57.4 | 43.8 [28.2–60.7] | 91.9 [82.5–96.5] | 35.69 | 5.43 | 0.61 | 0.38 | 0.94 |
| > 65.2 | 34.4 [20.4–51.7] | 95.2 [86.7–98.7] | 29.54 | 7.10 | 0.69 | 0.44 | 0.93 |
| > 78.0 | 25 [13.3–42.1] | 96.8 [89.0–99.4] | 21.77 | 7.74 | 0.78 | 0.46 | 0.92 |
| > 84.1 | 18.8 [8.9–35.3] | 98.4 [91.4–99.9] | 17.14 | 11.65 | 0.83 | 0.56 | 0.92 |
| > 115.4 | 6.3 [1.1–20.2] | 98.4 [91.4–99.9] | 4.64 | 3.88 | 0.95 | 0.30 | 0.90 |
| > 126.3 | 3.1 [0.2–15.7] | 100 [94.2–100] | 3.125 | | 0.97 | 1.00 | 0.90 |

CI: confidence interval; LR +: positive likelihood ratio; LR-: negative likelihood ratio; NPV: negative predictive value; PPV: positive predictive value.

and because of the use of numerous couples of PRC-PAC assays. These issues have led to an underestimation of PA diagnosis. In this context, 24UA excretion may be a diagnostic aid for PA diagnosis when hyperaldosteronism is unclear [32].

Numerous methods are currently available to measure PRC, PAC, or even 24UA using radio-immuno-assay, electro-chemiluminescent assay, or LC-MS/MS. The Lumipulse® (Fujirebio®, Tokyo, Japan) is an analyzer recently developed for these three measurements. However, given the high inter-method variability, specific cut-off values must be defined for each analyzer. Thus, the aim of this study was therefore 1) to analytically validate PRC, PAC, and 24UA measurements using Lumipulse®, and 2) to suggest criteria for predicting PA diagnosis based on ARR, PAC, and 24UA cutoff values.

Lumipulse® required a plasma volume of 40 μL and 50 μL to measure PAC and PRC, respectively. These volumes are smaller than those required by most immuno-assays and LC-MS/MS methods. Moreover, the possibility of measuring both PRC and PAC using the same EDTA-K3 plasma sample simplifies laboratory workflow by using the same aliquots. Furthermore, the results were obtained in less than 1 hour for each measurement without radioactivity.

Analytical evaluation showed excellent performances for PRC, PAC, and 24UA measurements with a correct imprecision and analytical efficiency with the number of samples for PAC and PRC (n = 196) and for 24UA (n = 201). Our results are in accordance with the three previous evaluations of these three parameters using Lumipulse® [14,15,33].

Our study showed a strong correlation between PRC measured with Lumipulse® and the reference Liaison XL® ($R^2 = 0.99$), with a significant tendency to underestimate PRC with Lumipulse ® with a mean bias of –7.6 mIU/L. Coulon et al. evaluated the analytical performance of Lumipulse® for PAC and PRC (n = 107) in comparison to the immuno-assay analyzer Isys® (IDS, Boldon, UK) [14]. Interestingly, their PRC comparison with the Isys® assays (n = 79) also showed a significant underestimation of PRC with Lumipulse® with a mean bias of –15.4 mIU/L, as highlighted in our study.

Our data also highlighted a strong correlation between Lumipulse® and our reference method LC-MS/MS for PAC (R = 0.98). Vermue et al. compared the PAC Lumipulse® assay with the reference LC-MS/MS in both routine (n = 39) and AVS (n = 133 AVS from 50 PA patients) [15]. They showed that PAC CV imprecision exceeded 2.4%. They also showed a strong correlation between Lumipulse® and LC-MS/MS PAC, with a slight tendency to overestimate with Lumipulse® with a smaller number of peripheral samples (n = 39), whereas our data showed a slight tendency to underestimate PAC with Lumipulse® with a larger number of samples (n = 195 peripheral samples). Ono et al. recently compared their own LC-MS/MS with PAC Lumipulse® (n = 344) with $PAC_{Lumipulse} = 0.962 \times PAC_{LC\text{-}MS/MS} - 1.2$ with $R^2 = 0.994$ [16]. Other recent studies have also confirmed the strong correlations between Lumipulse® and their LC-MS/MS for PAC in peripheral or AVS samples [17,18]. Finally, Teruyama et al. evaluated Lumipulse® using samples from 52 PA and 23 EH patients [33]. They highlighted an excellent correlation between Lumipulse® PRC and PAC and their previous respective radio-immuno-assay. Moreover, they found excellent results with the certified reference material of aldosterone NMIJ CRM 6402-a with a recovery between 99% and 104%. Their comparison with LC-MS/MS PAC also showed an excellent correlation ($R^2 = 0.996$, n = 41). Finally, all Lumipulse® PAC comparisons to the reference LC-MS/MS methods exhibited an excellent correlation, contrary to numerous other immuno-assays, probably thanks to the use of a sandwich method instead of the routine competition immuno-assay [9,34].

Nevertheless, none of these studies indicates specific cut-off values for ARR and PAC for PA diagnosis prediction using both PRC and PAC with Lumipulse®. In our cohort, the optimal cut-off for ARR was 35 pmol/mIU, which yielded a sensitivity of 95.5% and specificity of 67.8% (PPV: 24.8%; NPV: 99.3%) for PA detection. A 100% sensitivity would imply a lower ARR cut off value of 19.7 pmol/mIU that would reach only 37% specificity and a very low positive Likelihood ration of 1.6. With the same cohort, the ARR cut-off with our reference methods was 40.5 pmol/mIU (sensitivity: 97.7%; specificity: 76.3%; PPV: 31.4%; NPV:

99.7%) (S1A and S1B Fig), equivalent to those defined in the reference cohort (i.e., 46 pmol/mIU) [9]. In comparison, Teruyama et al. proposed a smaller ARR cut-off of 18 pmol/mIU (sensitivity: 96.2%; specificity: 78.3%) in a Japanese population with a smaller number of patients and without the discontinuation of RAAS-interfering drugs. International guidelines recommended the use of cut-off values for ARR ranging from 66 to 136 pmol/mIU [7]. In our analyses, these cut-off values would result in insufficient sensitivity of less than 60% for the PA detection (Table 3). As already stressed by numerous studies, each pair of PRC-PAC measurements requires corresponding ARR cut-off values for PA diagnosis prediction. With the electrochemiluminescence assay for PRC and PAC pairs, the suggested ARR cut-off values were concordant with those defined in our study: 31 pmol/mIU with Isys® (IDS®) [35] and 57.2 pmol/mIU with Liaison XL® (Diasorin®) [11].

Regarding PAC, our study exhibited a cut-off value of 260 pmol/L with Lumipulse® and yielded a sensitivity of 81.8%, specificity of 67.8%, PPV of 22%, and NPV of 97.1%; this is consistent with the PAC LC-MS/MS with a cut-off value of 255 pmol/L (S1C and S1D Fig). In a Japanese cohort of hypertensive patients, Ono et al. reached 85% sensitivity and 75.8% specificity with a cut off value of 277 pmol/L for PAC to distinguish APA from non-APA (both BAH and EH patients) [16]. In our study, such cut off would slightly decrease sensitivity of PA detection (72.7%) without improvement of specificity, because of the choice to diagnose only APA, with a potent highest proportion of KCNJ5 mutated APA as commonly observed in Asian cohorts. [36,37]. In our cohort a PAC below 178 pmol/L could exclude PA diagnosis.

In addition to ARR and PAC, PA diagnosis prediction could be improved with 24UA [32], as the excretion in urine over 24 hours integrates the daily fluctuations of aldosterone secretion and is independent of circadian variations [38]. Given the high inter-assay variability of urinary aldosterone measurements, it is necessary to determine a specific assay cut-off value [39]. Our study showed a strong correlation between 24UA measured with Lumipulse® and the reference LC-MS/MS assay with $R^2 = 0.88$. Finally, the 24UA cut-off value determined on Lumipulse® was 49 nmol/24h (sensitivity: 62.5%; specificity: 83.87%; PPV, 30.1%; NPV 95.3%), whereas it was 58 nmol/24h (sensitivity: 62.5%; specificity: 77.42%; PPV 23.5%; NPV 94.9%) for our reference LC-MS/MS (S1E and S1F Fig), similar to our reference cut-off value of 65 nmol/24h to distinguish PA from EH patients [40,41]. In our cohort, a 24UA below 11.9 nmol/24h could exclude PA diagnosis.

As expected, Lumipulse® 24UA had lower values than LC-MS/MS methods as already showed in previous studies [42,43]. These lower values are observed with most imuno-assays than with LC-MS/MS. This discrepancy, up to 30%, could be related to the diversity of acid hydrolysis conditions rather than discrepancies in calibrators choice [39,44]. A few studies have reported a reference interval for 24UA using immune assay methods. Teruyama et al. was the only previous study to evaluate 24UA with Lumipulse® based on only 18 samples but without interpretative criteria [33]. Our findings are in accordance with those of Yin et al. using the electrochemiluminescence assay method (Diasorin®) in healthy individuals (n = 114) with an upper range of 47 nmol/24h [45], Bekkach et al. with 37 nmol/24h using LC-MS/MS [43], and Kaneko et al. with 46 nmol/24h using the electrochemiluminescence assay [46].

## Limitations

Our study was limited by its small sample size and retrospective monocentric nature. Further prospective studies with larger sample sizes from multiple centers are required to assess more powerful diagnostic criteria for PA diagnosis prediction with ARR, PAC, and 24UA using Lumipulse®, possibly with sex-related cut-offs, and to also define the cut-off value for PA confirmatory tests.

## Conclusion

Our study demonstrates the accuracy and robustness of PRC, PAC, and 24UA measurements taken using Lumipulse® and establishes appropriate cut-off values for predicting PA diagnosis in a hypertensive population: ARR > 35 pmol/mIU, PAC > 260 pmol/L, and 24UA > 49 nmol/24h.

## Supporting information

**S1 Table. Analytical performances of plasma renin (PRC), plasma aldosterone (PAC), and urinary aldosterone (24UA) on Lumipulse® (Fujirebio®).** Results are expressed as mean +/- standard deviation; coefficient of variation (CV).
(DOCX)

**S1 Fig. Primary aldosteronism cutoff definition using Liaison XL and liquid chromatography coupled to tandem mass spectrometry (LC-MS/MS) for aldosterone to renin ratio (ARR): A) PA and EH ARR distribution, B) ARR ROC curve; for plasma aldosterone concentration (PAC), C) PA and EH PAC distribution, D) PAC ROC curve; for 24-hour urine aldosterone excretion (24UA), E) PA and EH 24UA distribution, F) 24UA ROC curve.**
(TIFF)

## Acknowledgments

Special acknowledgement to Els Huyck, Salima Alkhalfioui, and Nathalie Le Bastard for their help with Lumipulse® and the sample measurements, and to Dr Marie-Agnès Durey-Dragon, for gestion of ethics registration of "Samples requalification" on the behalf of the task force "Post-analysis use of biological samples" from Laboratory of Hopital européen Georges Pompidou.

## Author contributions

**Conceptualization:** Stéphanie Baron.

**Data curation:** Maude Laney, Appoline Nabi-Bourgois, Julien Mallart-Riancho, Simon Travers, Mouna Chebbi, Aurore Kling, Laurence Amar, Stéphanie Baron.

**Formal analysis:** Maude Laney, Appoline Nabi-Bourgois, Stéphanie Baron.

**Funding acquisition:** Stéphanie Baron.

**Investigation:** Maude Laney, Appoline Nabi-Bourgois, Julien Mallart-Riancho, Simon Travers, Mouna Chebbi, Aurore Kling, Laurence Amar, Stéphanie Baron.

**Methodology:** Stéphanie Baron.

**Project administration:** Stéphanie Baron.

**Resources:** Maude Laney, Appoline Nabi-Bourgois, Julien Mallart-Riancho, Simon Travers.

**Supervision:** Stéphanie Baron.

**Validation:** Laurence Amar, Stéphanie Baron.

**Writing – original draft:** Maude Laney.

**Writing – review & editing:** Julien Mallart-Riancho, Simon Travers, Laurence Amar, Stéphanie Baron.

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
