## [Decision Letter · Decision Letter 0]

22 Dec 2024

PONE-D-24-48138Criteria for diagnosing primary aldosteronism using Lumipulse® for renin and aldosterone measurementsPLOS ONE

Dear Dr. Baron,

Thank you for submitting your manuscript to PLOS ONE. After careful consideration, we feel that it has merit but does not fully meet PLOS ONE’s publication criteria as it currently stands. Therefore, we invite you to submit a revised version of the manuscript that addresses the points raised during the review process. I contacted many potential reviewers, but only I received a comment from one reviewer. Reading this comment, it seems that the methods need to be improved. Please revise it in a positive manner.

We look forward to receiving your revised manuscript.

Kind regards,

Etsuro Ito, Ph.D.

Academic Editor

PLOS ONE

Journal Requirements:

2. Thank you for stating the following financial disclosure: Fujirebio Europe sponsored this study, but did not participate in data analysis nor in writing the manuscript. 

Reviewers' comments:

Reviewer's Responses to Questions

**Comments to the Author**

1. Is the manuscript technically sound, and do the data support the conclusions?

Reviewer #1: Partly

2. Has the statistical analysis been performed appropriately and rigorously? 

Reviewer #1: I Don't Know

3. Have the authors made all data underlying the findings in their manuscript fully available?

Reviewer #1: No

4. Is the manuscript presented in an intelligible fashion and written in standard English?

Reviewer #1: Yes

5. Review Comments to the Author

Reviewer #1: This article shows the results of the comparison of measurements of PRC, PAC, and 24-h urine aldosterone excretion (24UA) between Lumipulse® and established reference methods and CVs that indicate measurement accuracy. These results are consistent with those of previous reports. However, a larger sample size may be a strong point. Additionally, the cutoff values for ARR (PAC/PRC), PAC, and 24UA in measurements by Lumipulse®, which are optimal for distinguishing between PA and EH diagnosed by confirmatory tests using the gold standard measurement methods, are provided. These data are valuable.

However, this manuscript has several problems with its main thesis and title that require revisions.

Major issues

Lines 43–45, 277–278, and 358–359

The authors indicate optimal (highest Youden index) cutoffs of ARR, PAC, and 24UA in measurements by Lumipulse® for distinguishing between PA and EH diagnosed by confirmatory tests; however, these are not equivalent to being appropriate cutoffs as screening criteria. Screening aims to identify various suspected patients and is not solely determined by the rate of concordance with the final diagnosis. The cutoff values presented by the authors are just for diagnostic prediction. It would be desirable not to state directly that “these are screening criteria,” but to limit the statement to “the cutoff shows sensitivity of ~~% and specificity of ~~% for PA screening,” as in other reference literatures.

The establishment of screening criteria requires consideration of cost-benefit, an acceptable overlook rate, and the boundary between PA and EH that should be set as a borderline between treatment strategies, considering not only confirmatory test values but also complications, disease types, and pathological findings.

Moreover, if the initial screening criteria differ among measurement kits, it may cause confusion in screening by general practitioners and field clinics, leading to the avoidance of screening itself. Although the screening criteria for each test kit should be considered, it would be preferable to initially establish uniform screening criteria based on standard measurement methods (LC-MS/MS) and adjust the criteria to match those for each kit. If the screening criteria are defined based on the sensitivity and specificity of the highest Youden index for each kit, the combination of sensitivity and specificity at the cutoff with the highest Youden index will differ, and the characteristics of each screening criterion will be different.

However, some aspects of the present results suggest that lowering the screening criteria would be desirable and such a proposal should be considered.

Lines 1–2

The present study validates the accuracy of Lumipulse® measurement, and shows cutoff values with Lumipulse® to predict PA diagnosis by conventional criteria in cases at the author’s institution, but not the diagnostic criteria for PA itself. Therefore, the study’s current title was inappropriate for the content. The title should coincide more with the content of the manuscript, such as “Measurement accuracy and cutoffs for predicting primary aldosteronism diagnosis using Lumipulse® for renin and aldosterone measurements.”

Lines 201–202

Of the patients with EH, 22% receive potassium supplementation. What would have caused hypokalemia if not PA?

Lines 130–132 and 136–137

It is stated that only samples from patients without drugs interfering with the RAAS were used. What are the drugs that you defined as precisely interfering with the RAAS?

Lines 184–185

While Lumipulse® and LC-MS/MS provided similar values for PAC, why does Lumipulse® show smaller values for urinary aldosterone than LC-MS/MS?

Minor issues

Line 104

Is the unit of ARR not pmol/mIU?

Lines 192–193

Why did only 11 of the 18 unilateral PA patients undergo surgery?

Lines 203–204, Tables

The letter “p” is inconsistently expressed in lower and upper case. The significant digits of the p-values are not consistent throughout the text and tables and vary from <0.05–<0.0001.

Lines 329–333

The authors presented a cutoff of 260 pmol/L as the boundary between EH and PA, whereas the value of 277 pmol/L presented by Ono et al. was the cutoff used to distinguish APA from IHA and EH. Were these results consistent?

6. PLOS authors have the option to publish the peer review history of their article (what does this mean? ). If published, this will include your full peer review and any attached files.

**Do you want your identity to be public for this peer review?** For information about this choice, including consent withdrawal, please see our Privacy Policy .

Reviewer #1: No

---

## [Author Response · Author response to Decision Letter 0]

7 Jan 2025

We would like to thank the reviewer for the careful review of our manuscript.

As detailed below and in the revised version of the manuscript, we have considered all your suggestions that greatly contributed to improve our manuscript. Below are our responses to your suggestions. We underlined in yellow the part of the manuscript that have been added or rephrased; we barred the part or sentences that have been deleted. All these changes have been highlighted using “Track Changes” option in Microsoft Word.

In addition, supplemental figure 1 has been reuploaded with correction of unit defined in the figure for 24UA expression with LC-MS/MS (nmol/24h instead of µg/d).

Of course, we would be more than glad to answer any additional questions or comments.

Herein, we provide a point-by-point response to the reviewers’ comments.

Journal Requirements:

As requested, we checked that the revised version of our manuscript meets style’s requirement for Plos One manuscript.

2. Thank you for stating the following financial disclosure: Fujirebio Europe sponsored this study, but did not participate in data analysis nor in writing the manuscript.

As required, we stated the role of the funder in our study in the conflict-of-interest area of the manuscript. In this way, as suggested, we added lines 389-390 the sentence “This study was sponsored by Fujirebio Europe® but the funders had no role in study design, data collection and analysis, decision to publish, or preparation of the manuscript.”

The cover letter also mentioned the amended role of the Funder as requested with the sentence: “This study was sponsored by Fujirebio Europe, but the funders had no role in study design, data collection and analysis, decision to publish, or preparation of the manuscript”.

Reviewers' comments:

Reviewer's Responses to Questions

Comments to the Author

1. Is the manuscript technically sound, and do the data support the conclusions?

Reviewer #1: Partly

2. Has the statistical analysis been performed appropriately and rigorously?

Reviewer #1: I Don't Know

3. Have the authors made all data underlying the findings in their manuscript fully available?

The PLOS Data policy requires authors to make all data underlying the findings described in their manuscript fully available without restriction, with rare exception (please refer to the Data Availability Statement in the manuscript PDF file). The data should be provided as part of the manuscript or its supporting information or deposited to a public repository. For example, in addition to summary statistics, the data points behind means, medians and variance measures should be available. If there are restrictions on publicly sharing data—e.g. participant privacy or use of data from a third party—those must be specified.

Reviewer #1: No

4. Is the manuscript presented in an intelligible fashion and written in standard English?

Reviewer #1: Yes

5. Review Comments to the Author

Reviewer #1: This article shows the results of the comparison of measurements of PRC, PAC, and 24-h urine aldosterone excretion (24UA) between Lumipulse® and established reference methods and CVs that indicate measurement accuracy. These results are consistent with those of previous reports. However, a larger sample size may be a strong point. Additionally, the cutoff values for ARR (PAC/PRC), PAC, and 24UA in measurements by Lumipulse®, which are optimal for distinguishing between PA and EH diagnosed by confirmatory tests using the gold standard measurement methods, are provided. These data are valuable.

We thank the reviewer for his carefully reading of our manuscript.

However, this manuscript has several problems with its main thesis and title that require revisions.

Major issues

Lines 43–45, 277–278, and 358–359

The authors indicate optimal (highest Youden index) cutoffs of ARR, PAC, and 24UA in measurements by Lumipulse® for distinguishing between PA and EH diagnosed by confirmatory tests; however, these are not equivalent to being appropriate cutoffs as screening criteria. Screening aims to identify various suspected patients and is not solely determined by the rate of concordance with the final diagnosis. The cutoff values presented by the authors are just for diagnostic prediction. It would be desirable not to state directly that “these are screening criteria,” but to limit the statement to “the cutoff shows sensitivity of ~~% and specificity of ~~% for PA screening,” as in other reference literatures.

The establishment of screening criteria requires consideration of cost-benefit, an acceptable overlook rate, and the boundary between PA and EH that should be set as a borderline between treatment strategies, considering not only confirmatory test values but also complications, disease types, and pathological findings.

We’d like to thank the reviewer for this suggestion. Indeed, screening criteria must consider both sensitivity of the detection, and cost-benefit consideration. In our study, we considered 3 cut-off values for 3 criteria that must be considered for PA diagnosis, with different sensitivity and specificity performances. Our criteria were defined in accordance with the highest Youden index for each parameter to define an acceptable compromise between sensitivity and specificity.

Indeed, if we focused on sensitivity only, as requested by screening tests, our suggested cutoff values would have been lower. A 100% sensitivity would have led to the use of 19.7 pmol/mIU as cut off value for ARR with 37.3% specificity; 178 pmol/L for 100% sensitivity PAC with 39.0% specificity; and 11.9 nmol/24H for 24UA with 4.8% specificity.

Regarding specifically ARR, our highest Youden index cut off value

Regarding the usual screening criteria ARR. In our study, highest Youden index already reached 95.5% sensitivity, with 67.8% specificity. If we choose to increase sensitivity to 97.7% our cut off value is not significantly different with 31.4 pmol/mIU with 57.6% specificity. If we reached 100% sensitivity, cut off ARR would be 19.7 pmol/mIU with only 37.3% specificity. In this way, regarding ARR our cutoff value with highest Youden index did not differ from cut off value focused on optimal sensitivity. (Table 3)

In this way, we have modified sentence line 43-45 as below to clarify the use of our suggested cutoff values:

“We propose cut-off values of 35 pmol/mIU as the ARR, 260 pmol/L for PAC, and 49 nmol/24h for 24UA for PA diagnosis.

In this way, we also changed the word screening throughout the manuscript to improve clarity of the manuscript.

- Line 74: screening turned into diagnosis

- Line 192: screening turned into diagnosis

- Line 235: screening turned into detection

- Line 243: screening turned into detection

- Line 248: screening turned into detection

- Line 251: screening turned into detection

- Line 257: screening turned into detection

- Line 264: screening turned into detection

- Line 271: screening turned into detection

- Line 275: screening turned into detection

- Line 330: screening turned into detection

- Line 339: screening test turned into PA detection

- Line 341: screening turned into diagnosis

- Line 354: screening PA turned into PA diagnosis

- Line 382: screening PA turned into PA diagnosis

Moreover, if the initial screening criteria differ among measurement kits, it may cause confusion in screening by general practitioners and field clinics, leading to the avoidance of screening itself. Although the screening criteria for each test kit should be considered, it would be preferable to initially establish uniform screening criteria based on standard measurement methods (LC-MS/MS) and adjust the criteria to match those for each kit.

As underlined by reviewer, each combination of aldosterone and renin measurements could lead to potent different criteria to diagnose PA. Nonetheless, if LC-MS/MS measurement is considered as the gold standard for aldosterone (plasma and urine) measurement, there is no gold standard for renin measurement even though there is international standardization for aldosterone. Regarding aldosterone measurement, even though LC-MS/MS methods become more and more widespread, plasma aldosterone concentration could differ up to 40% depending on method and calibration (Fanelli et al. 2022). International harmonization of those measurements is yet not achieved. Regarding 24UA, there is currently no consensus regarding cut off value whatever the method in its use in PA diagnosis.

If the screening criteria are defined based on the sensitivity and specificity of the highest Youden index for each kit, the combination of sensitivity and specificity at the cutoff with the highest Youden index will differ, and the characteristics of each screening criterion will be different. However, some aspects of the present results suggest that lowering the screening criteria would be desirable and such a proposal should be considered.

We’d like to thank the reviewer for this suggestion. Indeed, we choose to use criteria corresponding to the highest Youden index for each parameter.

Regarding the usual screening criteria ARR, in our study, highest Youden index reached 95.5% sensitivity, with 67.8% specificity. If we choose to increase sensitivity to 97.7% our cut off value is not significantly different with 31.4 pmol/mIU with 57.6% specificity. If we reached 100% sensitivity, cut off ARR would be 19.7 pmol/mIU with only 37.3% specificity. In this context, our cutoff value for ARR with highest Youden index would not differ from cut off value focused on optimal sensitivity (Table 3).

However, regarding PAC, our highest Youden index based cutoff value reach 81.8% sensitivity and 67.8% specificity. If we focused on sensitivity in case of a screening test, to reach 90% sensitivity, PAC cut off value would be 197 pmol/L with only 47.5% specificity. On the other side, if we focused on specificity, 90% specificity would be reached with a cutoff value of 450 pmol/L with only 47.7% sensitivity (Table 4).

Moreover, using the same strategy for 24UA cut off value, our suggested cut off value of 49 nmol/L reached 62.5% sensitivity with 83.9% specificity. If we focused on sensitivity, 90% would be reached with a cutoff value of 22.8 nmol/L24h with only 22.6% specificity; if we focused on specificity, 90% specificity would be reached with cut off value of 57 nmol/24H with only 43.8 sensitivity (Table 5).

As suggested by reviewer, we added this information for each criterion.

- Line 244: A cut off value of 31.4 pmol/mIU would reach 97.7% sensitivity and 57.6% specificity. To reach 100% sensitivity, corresponding cut off value for ARR would be 19.7 pmol/mIU with 37.3% specificity.

- Line 258: To reach 90% sensitivity, PAC cut off value would be 197 pmol/L with 47.5% specificity; while to reach 90% specificity the cut off value would be 450 pmol/L with 47.7% sensitivity.

- Line 272: A 90% sensitivity would be reached with a cutoff value of 22.8 nmol/L24h with 22.6% specificity; a 90% specificity would be reached with cut off value of 57 nmol/24h with 43.8% sensitivity.

- Line 330: A 100% sensitivity would imply a lower ARR cut off value of 19.7 pmol/mIU that would reach only 37% specificity and a very low positive Likelihood ration of 1.6

- Line 353: In our cohort a PAC below 178 pmol/L could exclude PA diagnosis.

- Line 363: In our cohort, a 24UA below 11.9 nmol/24h could exclude PA diagnosis

Lines 1–2

The present study validates the accuracy of Lumipulse® measurement and shows cutoff values with Lumipulse® to predict PA diagnosis by conventional criteria in cases at the author’s institution, but not the diagnostic criteria for PA itself. Therefore, the study’s current title was inappropriate for the content. The title should coincide more with the content of the manuscript, such as “Measurement accuracy and cutoffs for predicting primary aldosteronism diagnosis using Lumipulse® for renin and aldosterone measurements.”

As suggested by the reviewer, we can improve the readability of our study by changing the title of our manuscript using “Measurement accuracy and cutoffs for predicting primary aldosteronism diagnosis using Lumipulse® for renin and aldosterone measurements”.

Lines 201–202

Of the patients with EH, 22% receive potassium supplementation. What would have caused hypokalemia if not PA?

We thank the reviewer for this remark. Indeed, 22% of our EH patients received potassium supplementation, while 75% of PA patients did. Nonetheless, EH patients did not exhibit hypokalemia, while PA patients exhibited significantly decreased kalemia despite potassium supplementation. Moreover, as underlined by potassium excretion, EH patients exhibited significantly lower urinary potassium excretion than PA patients.

In our reference center, potassium supplementation was systematically prescribed when patients exhibited a history of acute or chronical, spontaneous or diuretic-induced hypokalemia prior to primary aldosteronism screening (in accordance with International Guidelines (Funder, 2016).

In this way, we added line 208:

“Potassium supplementation was systematically prescribed when patients exhibited any history of hypokalemia in accordance with guidelines [7].”

Lines 130–132 and 136–137

It is stated that only samples from patients without drugs interfering with the RAAS were used. What are the drugs that you defined as precisely interfering with the RAAS?

As noticed by reviewer, RAAS interfering treatment were discontinued before blood sampling if used. To clarify this point, sentence beginning line 132 has been added for plasma criteria:

“Beta-blockers, angiotensin II type 1 receptor blockers, angiotensin-converting enzyme inhibitors and diuretics were discontinued 2 weeks before blood collection; mineralocorticoid receptor antagonists and renin inhibitors 6 weeks before. Calcium antagonists, α-blockers, and centrally acting antihypertensive drugs were prescribed to patients with severe hypertension if necessary.”

In the same way, sentence beginning line 141 has been changed in urinary criteria section:

“For PA diagnosis, only samples from patients without drugs interfering with RAAS (as described previously) were included (i.e., n=32 PA and n=62 EH).”

Lines 184–185

While Lumipulse® and LC-MS/MS provided similar values for PAC, why does Lumipulse® show smaller values for urinary aldosterone than LC-MS/MS?

We thank the reviewer for this interesting question. Indeed, we do observe lower urinary aldosterone concentrations with Lumipulse® than with our LC-MS/MS. Fujirebio method is standardized against CRM 6402-a while our LC-MS/MS is standardized using Cerilliant® certified reference material.

However, there are very few studies focusing on urinary aldo

---

## [Decision Letter · Decision Letter 1]

24 Jan 2025

PONE-D-24-48138R1Measurement accuracy and cutoffs for predicting primary aldosteronism diagnosis using Lumipulse® for renin and aldosterone measurementsPLOS ONE

Dear Dr. Baron,

Thank you for submitting your manuscript to PLOS ONE. After careful consideration, we feel that it has merit but does not fully meet PLOS ONE’s publication criteria as it currently stands. Therefore, we invite you to submit a revised version of the manuscript that addresses the points raised during the review process. The comments from the reviewer seem very minor.Please submit your revised manuscript soon.

We look forward to receiving your revised manuscript.

Kind regards,

Etsuro Ito, Ph.D.

Academic Editor

PLOS ONE

Journal Requirements:

Reviewers' comments:

Reviewer's Responses to Questions

**Comments to the Author**

1. If the authors have adequately addressed your comments raised in a previous round of review and you feel that this manuscript is now acceptable for publication, you may indicate that here to bypass the “Comments to the Author” section, enter your conflict of interest statement in the “Confidential to Editor” section, and submit your "Accept" recommendation.

Reviewer #1: (No Response)

2. Is the manuscript technically sound, and do the data support the conclusions?

Reviewer #1: Partly

3. Has the statistical analysis been performed appropriately and rigorously? 

Reviewer #1: I Don't Know

4. Have the authors made all data underlying the findings in their manuscript fully available?

Reviewer #1: No

5. Is the manuscript presented in an intelligible fashion and written in standard English?

Reviewer #1: Yes

6. Review Comments to the Author

Reviewer #1: Most of the previous suggestions have been appropriately modified in the revised manuscript. However, it should be noted that the predictive ability for the diagnosis of PA (for which criteria have already been established) at the cutoff values of ARR, PAC, and 24UA presented by the authors does not constitute the diagnostic criteria itself, even though it may be a resource for developing screening criteria.

Lines 43–45, 288–290, 379–381

The authors do not present the PA diagnostic criteria per se, but rather cutoff values which predict PA diagnosis. Therefore, the following revisions are required:

“We propose cut-off values of 35 pmol/mIU for ARR, 260 pmol/L for PAC, and 49 nmol/24h for 24UA for predicting PA diagnosis.” (Lines 43–45)

“Thus, the aim of this study was therefore 1) to analytically validate PRC, PAC, and 24UA measurements using Lumipulse®, and 2) to suggest criteria for predicting PA diagnosis based on ARR, PAC, and 24UA cutoff values.” (Lines 288–290)

“Our study demonstrates the accuracy and robustness of PRC, PAC, and 24UA measurements taken using Lumipulse® and establishes appropriate cut-off values for predicting PA diagnosis in a hypertensive population: ARR > 35 pmol/mIU, PAC > 260 pmol/mL, and 24UA > 49 nmol/24h.” (Lines 379–381)

In addition, similar revisions are needed in other sentences which refer to “PA diagnosis” (Lines 47, 70, 74, 189, 325, 339, 352, and 376).

Lines 69–71

Some other studies recently reported on the relationship of PAC between Lumipulse and LC-MS/MS in peripheral or AVS samples (Taki et al. 2024, Nakai et al. 2024). These studies also reported a strong correlation, which would support the present results.

Taki et al. (2024) Scientific Reports 14:26040

Nakai et al. (2024) Endocr J 71(5):461-469

Comment only:

Lines 363–

The description in the revised manuscript regarding the reasons why urinary aldosterone measurements differ between the immunoassay and LC-MS/MS methods is reasonable and interesting.

7. PLOS authors have the option to publish the peer review history of their article (what does this mean? ). If published, this will include your full peer review and any attached files.

**Do you want your identity to be public for this peer review?** For information about this choice, including consent withdrawal, please see our Privacy Policy .

Reviewer #1: No

---

## [Author Response · Author response to Decision Letter 1]

27 Jan 2025

We would like to thank the reviewer for the detailed and constructive feedback provided on our manuscript. Below, we address each comment thoroughly and describe the changes made in the revised version of the manuscript. All revisions are clearly indicated in the manuscript using the "Track Changes" feature, with additions and rephrased sections highlighted in yellow, and deletions struck through for clarity.

Should there be any additional questions or comments, we remain at your full disposal to provide further clarification.

Herein, we provide a point-by-point response to the reviewers’ comments.

Journal Requirements Compliance:

Please review your reference list to ensure that it is complete and correct. If you have cited papers that have been retracted, please include the rationale for doing so in the manuscript text or remove these references and replace them with relevant current references. Any changes to the reference list should be mentioned in the rebuttal letter that accompanies your revised manuscript. If you need to cite a retracted article, indicate the article’s retracted status in the References list and also include a citation and full reference for the retraction notice.

We have carefully reviewed our reference list to ensure it is complete and up to date.

Reviewer's Comments and Author Responses

Reviewer #1: Most of the previous suggestions have been appropriately modified in the revised manuscript. However, it should be noted that the predictive ability for the diagnosis of PA (for which criteria have already been established) at the cutoff values of ARR, PAC, and 24UA presented by the authors does not constitute the diagnostic criteria itself, even though it may be a resource for developing screening criteria.

Lines 43–45, 288–290, 379–381

The authors do not present the PA diagnostic criteria per se, but rather cutoff values which predict PA diagnosis. Therefore, the following revisions are required:

“We propose cut-off values of 35 pmol/mIU for ARR, 260 pmol/L for PAC, and 49 nmol/24h for 24UA for predicting PA diagnosis.” (Lines 43–45)

“Thus, the aim of this study was therefore 1) to analytically validate PRC, PAC, and 24UA measurements using Lumipulse®, and 2) to suggest criteria for predicting PA diagnosis based on ARR, PAC, and 24UA cutoff values.” (Lines 288–290)

“Our study demonstrates the accuracy and robustness of PRC, PAC, and 24UA measurements taken using Lumipulse® and establishes appropriate cut-off values for predicting PA diagnosis in a hypertensive population: ARR > 35 pmol/mIU, PAC > 260 pmol/mL, and 24UA > 49 nmol/24h.” (Lines 379–381)

In addition, similar revisions are needed in other sentences which refer to “PA diagnosis” (Lines 47, 70, 74, 189, 325, 339, 352, and 376).

We appreciate the reviewer’s suggestion to distinguish between “diagnosis criteria” and “diagnostic prediction”. As recommended, we changed sentences mentioned above.

Moreover, throughout the manuscript, the term "diagnosis" has been replaced by "prediction" as appropriate. In this way, we have modified sentences (Lines 47, 70, 74, 189, 325, 339, 352, and 376) mentioned by the reviewer to clarify this distinction as described:

Line 47: Lumipulse® (Fujirebio®) is a reliable alternative for measuring PRC, PAC, and 24UA for predicting PA diagnosis with the use of corresponding cut-off values for each criterion.

Line 70: A few studies have evaluated this method [14-18], although none of them defined specific cut-off values for PA prediction using Lumipulse® renin and aldosterone measurements.

Line 73: The aim of this study was therefore 1) to compare Lumipulse® with the previously validated immuno-assay method for PRC and the gold standard LC-MS/MS method for PAC and 24UA, and 2) to define cut-off values for the prediction of PA diagnosis for ARR, PAC, and 24UA using Lumipulse®.

Line 192: Determining new cut-off values for predicting primary aldosteronism diagnosis

Line 331: Nevertheless, none of these studies indicates specific cut-off values for ARR and PAC for PA diagnosis prediction using both PRC and PAC with Lumipulse®.

Line 352: In addition to ARR and PAC, PA diagnosis prediction could be improved with 24UA [32], as the excretion in urine over 24 hours integrates the daily fluctuations of aldosterone secretion and is independent of circadian variations [38].

Line 382: Further prospective studies with larger sample sizes from multiple centers are required to assess more powerful diagnostic criteria for PA diagnosis prediction with ARR, PAC, and 24UA using Lumipulse®, possibly with sex-related cut-offs, and to also define the cut-off value for PA confirmatory tests.

Lines 69–71

Some other studies recently reported on the relationship of PAC between Lumipulse and LC-MS/MS in peripheral or AVS samples (Taki et al. 2024, Nakai et al. 2024). These studies also reported a strong correlation, which would support the present results.

We thank the reviewer for this suggestion. The studies by Taki et al. (2024) and Nakai et al. (2024) are indeed recent and highly relevant, as they confirm the robustness of PAC measurements using the Lumipulse® method in comparison with LC-MS/MS in both peripheral and AVS samples. We have added these references into introduction (line 70) and discussed their results (lines 319 to 321) to further support the validity of our findings.

Line 70: A few studies have evaluated this method [14-18], although none of them defined specific cut-off values for PA prediction using Lumipulse® renin and aldosterone measurements.

Line 319 – 321: Other recent studies have also confirmed the strong correlations between Lumipulse® and their LC-MS/MS for PAC in peripheral or AVS samples [17, 18].

Comment only:

Lines 363–

The description in the revised manuscript regarding the reasons why urinary aldosterone measurements differ between the immunoassay and LC-MS/MS methods is reasonable and interesting.

We’d like to thank the reviewer for this remark.

---

## [Editor Report · Decision Letter 2]

29 Jan 2025

Measurement accuracy and cutoffs for predicting primary aldosteronism diagnosis using Lumipulse® for renin and aldosterone measurements

PONE-D-24-48138R2

Dear Dr. Baron,

We’re pleased to inform you that your manuscript has been judged scientifically suitable for publication and will be formally accepted for publication once it meets all outstanding technical requirements.

Kind regards,

Etsuro Ito, Ph.D.

Academic Editor

PLOS ONE

---

## [Editor Report · Acceptance letter]

PONE-D-24-48138R2

PLOS ONE

Dear Dr. Baron,

I'm pleased to inform you that your manuscript has been deemed suitable for publication in PLOS ONE. Congratulations! Your manuscript is now being handed over to our production team.

Kind regards,

on behalf of

Prof. Etsuro Ito

Academic Editor

PLOS ONE